# A study of the impact of de-capacity policies on industry capacity utilization paths: Evidence from the Chinese steel industry

**Shixin Shi**[1]☯, **Hao Li**[2]☯*, **Hongsong Tang**[1], **Yang Liu**[3]

**1** Neijiang Normal University, Neijiang, Sichua, China, **2** Southwest Petroleum University, Chengdu, Sichuan, China, **3** Sichuan Changning Natural Gas Development Co., Ltd., Yibin, Sichuan, China

☯ These authors contributed equally to this work.

* 202121000581@stu.swpu.edu.cn

**Data Availability Statement:** The data for the explanatory variables, dependent variables, and control variables used in this paper can be

## Abstract

The issue of overcapacity has become an unavoidable challenge in the rapid development of nations, constraining economic progress, particularly within industries like steel, coal, and cement. This study, using the example of the Chinese steel industry in the context of supply-side structural reform, employs data envelopment analysis (DEA) models to measure capacity utilization, and ordinary least squares (OLS) models to investigate the impact of capacity reduction policies on the steel industry's capacity utilization pathways. The research findings indicate that capacity reduction policies have a significantly positive impact on the capacity utilization in the steel industry. They enhance capacity utilization through four pathways: "equipment optimization and upgrade", "enterprise mergers and restructuring", "technology innovation-driven", and "environmental protection regulations". Among these, "technology innovation-driven" and "environmental protection regulations" play predominant roles, while the effect of "international market expansion" on increasing capacity utilization in the steel industry is not significant. To ensure the sustained effectiveness of capacity reduction policies, the nation should continue to strengthen the "technology innovation-driven" and "environmental protection regulations" pathways. Additionally, it should activate the "national market expansion" pathway, fully exploring the potential for international cooperation to achieve improved capacity utilization in the steel industry.

## Introduction

Since the 1990s, overcapacity has consistently troubled China's economic development, especially in industries like steel, coal, and cement [1]. Prior to the 13th Five-Year Plan, China primarily attempted to address overcapacity by stimulating consumption to boost market demand, focusing on controlling the expansionary production behaviors of companies. However, with the influence of local government interventionism, the fundamental issue of overcapacity remained unresolved, and the capacity utilization in the industrial sector remained relatively low [2, 3]. After more than a decade of implementing measures to reduce overcapacity, it became evident that the earlier policies were not in line with China's transition to a new

**Funding:** This research was supported by the Social Science Foundation of China (17XJY014). The funders had no role in study design, data collection and analysis, decision to publish, or preparation of the manuscript.

**Competing interests:** The authors have declared that no competing interests exist.

economic paradigm focused on high-quality development. As a result, there arose a necessity to shift the approach for addressing overcapacity, with an emphasis on supply-side measures to reduce excess capacity.

In January 2016, the Chinese government formally introduced the Supply-Side Structural Reform, placing overcapacity reduction at the forefront of the reform's tasks. The following month, the "Opinions on Resolving Overcapacity and Achieving Sustainable Development in the Iron and Steel Industry" was released, which further clarified the goals of overcapacity reduction in the iron and steel sector and was accompanied by a series of policy measures aimed at comprehensively reducing overcapacity in the iron and steel industry at a macroeconomic level. The question arises: has the overcapacity reduction policy effectively increased the capacity utilization in the steel industry? What are the possible pathways through which it has an impact? In light of this, this article, set against the backdrop of the Supply-Side Structural Reform, investigates the impact of the de-capacity policies on the capacity utilization in the iron and steel industry, exploring its potential mechanisms. This research aims to enhance the theory of public management and public policy, providing decision-making insights for government departments seeking to optimize their de-capacity policies. Moreover, this study can serve as a reference for other countries facing similar overcapacity challenges.

The issue of overcapacity has become an inevitable problem in the rapid development of nations, constraining economic growth. Over the past few decades, a general decline in manufacturing capacity utilization in the United States due to weak investment has led to overcapacity issues [4]. Caballero, Hoshi, and Kashyap [5] found that in Japan, overcapacity problems were prevalent in industries with a higher proportion of zombie companies. Overcapacity issues are mostly a short-term phenomenon that accompanies economic crises, known as cyclical overcapacity [6, 7]. For developing countries, overcapacity problems occur more frequently compared to developed nations due to rapid changes in consumption patterns [1]. To address this problem, there has been a growing emphasis on supply-side management. After the 2008 financial crisis, the United States adopted a 'supply management' approach, focusing on policy operations and structural adjustments from the supply side, rather than just regulating monetary aggregates and demand-side adjustments [8]. As of the present time, clear signs of economic recovery have emerged due to these measures.

Addressing overcapacity poses significant challenges in both government economic and social management and remains a prominent topic in academic research. The existing literature has enriched research along various dimensions related to overcapacity. These dimensions include its characterization, measurement, formation mechanisms, resolution approaches, and effect assessment [9–12]. Within the context of China's supply-side structural reform, the steel industry has become the central focus for capacity reduction efforts, attracting significant attention in academic circles. The primary areas of research include the following.

Firstly, this section reviews the research on capacity utilization rate measurement and feature analysis in the iron and steel industry. The commonly employed approaches in academia for estimating capacity utilization include the peak method, the function method (including cost minimization and profit maximization standards), and the data envelopment analysis (DEA) method. These methods exhibit distinct differences in their assumptions and criteria for defining production capacity [13–16]. Feng et al. [17] used the comprehensive index method to analyze the degree of overcapacity in China's steel industry from 2000 to 2016, pointing out that China's steel industry had been in a state of excess capacity in recent years. Given the changing economic environment, this situation was expected to worsen further. Considering that China faces widespread industrial overcapacity, and recognizing the significant influence of non-market factors on capacity utilization, it is believed that the use of the data envelopment analysis method, which measures capacity utilization based on technological

production capabilities, aligns more with the Chinese context. This approach has been adopted by the majority of scholars for capacity utilization estimation in China [18–20]. Fukuyama et al. employed the data envelopment analysis (DEA) method to study the capacity utilization rate of some large iron and steel enterprises in China [19]. To address capacity utilization measurement while accounting for changing quasi-fixed inputs, Fukuyama et al. introduced a novel DEA-based method [20]. This approach aligns with the Chinese government's goal of reducing backward capacity over an intermediate timeframe [21]. In the Data Envelopment Analysis method, production capacity refers to the capacity when a firm's fixed capital is fully utilized, which can be understood as the technical definition of production capacity. However, this model only considers indicators favorable for socioeconomic development, i.e., expected output, and does not account for indicators unfavorable to socioeconomic development, i.e., non-desired output when designing output metrics. As a result, the technically defined production capacity calculated in this manner is evidently overestimated. The steel industry plays a significant role in China's national economic development, but it is not only an energy-intensive industry but also a highly polluting one. Therefore, the technically defined production capacity does not comprehensively and accurately represent the optimal production capacity of steel enterprises.

Secondly, the mechanism of overcapacity formation in the steel industry is examined. Mei and Chen [22] conducted vector autoregression (VAR) analysis on factors influencing China's steel overcapacity, including fixed asset investment, the growth rate of real estate construction area, steel export rate, steel industry concentration, iron ore prices, and the local government investment growth index. The results indicated that these six indices played significant roles in affecting overcapacity. Shadikhodjaev [23] suggested that government intervention and low market demand were contributing factors to overcapacity in the steel industry.

Third, the study explores overcapacity management mechanisms in the steel industry. Scholars have proposed effective strategies for addressing overcapacity in the steel industry, such as upgrading large and medium-sized advanced capacity to save energy, eliminating small and backward capacity, and enhancing efficient capacity [11, 20, 24]. Zhou and Yang [25] presented five recommendations to further promote emission reduction in China's steel industry, including phasing out outdated steel production capacity, enhancing the legal framework for emission control, strengthening government supervision, improving steel production technology and efficiency, and establishing a national carbon emission trading market for the steel industry.

Fourth, the evaluation of the impact of capacity removal in the steel industry is discussed. Currently, there is limited literature evaluating the effects of de-capacity policies in the context of supply-side structural reform. Tian et al. [26] employed the difference-in-difference method to investigate the impact of overcapacity reduction policies on the profitability of listed companies in the steel and coal industries. The results demonstrated a significant improvement in the return on equity for enterprises due to the overcapacity reduction policy. Wei et al. [27] developed an LEAP policy model to assess the performance of government-made industrial policies, revealing substantial reductions in $CO_2$ emissions if these policies are effectively implemented.

In general, there are still some shortcomings in the existing literature. In the context of supply-side structural reform, the existing literature has not considered the impact of non-desired output on the steel industry's capacity utilization rate. Additionally, there is limited literature assessing the effectiveness of de-capacity policies in the steel industry, with a lack of quantitative analysis regarding the pathways through which these policies affect the steel industry's capacity utilization rate. This article builds upon this foundation in two aspects: first, to measure the capacity utilization rate of the iron and steel industry using a data envelopment model

that considers non-desired output in the context of supply-side structural reform; second, to analyze the impact of capacity removal policies on the iron and steel industry from five aspects, including "equipment optimization and upgrading", "enterprise merger and reorganization", "technology innovation driving", "international market development", and "environmental protection regulation".

## Analysis of the path of the role of capacity utilization in the steel industry affected by the capacity removal policy

### Analytical framework

The ultimate goal of the de-capacity policy in the context of supply-side structural reform remains consistent with that of previous de-capacity policies. Its primary objective is to enhance the capacity utilization rate of industries facing excess capacity. So, what are the pathways through which the policy of capacity reduction in the steel industry, within the framework of supply-side structural reform, improves the steel industry's capacity utilization rate? This paper, based on the document "Opinions on Realizing Difficult Development of Iron and Steel Industry by Resolving Overcapacity", and in conjunction with specific practices for addressing overcapacity in China's iron and steel sector within the context of supply-side structural reform, explores the impact of capacity removal policies on the capacity utilization rate of the iron and steel industry through five avenues: "equipment optimization and upgrading", "enterprise merger and reorganization", "technology innovation driving", "international market development", and "environmental protection regulation" [28, 29].

From the perspective of the capacity utilization measurement model (data envelopment model), the factors influencing capacity utilization are the input and output indices of the model [30]. In essence, it reflects the ability and efforts of the iron and steel industry in its pursuit of high-quality development. It provides a comprehensive view of the impact of technological progress on economic development, resource utilization, and the ecological environment, aligning with the demands of the contemporary era for high-quality development.

As a result, a logical framework for analyzing the impact of capacity utilization in the steel industry through capacity removal policies has been established, as illustrated in Fig 1. Capacity removal policies have a "resource-saving effect", an "economic high-yield effect", and a "pollution reduction effect" on the steel industry through five pathways: "equipment

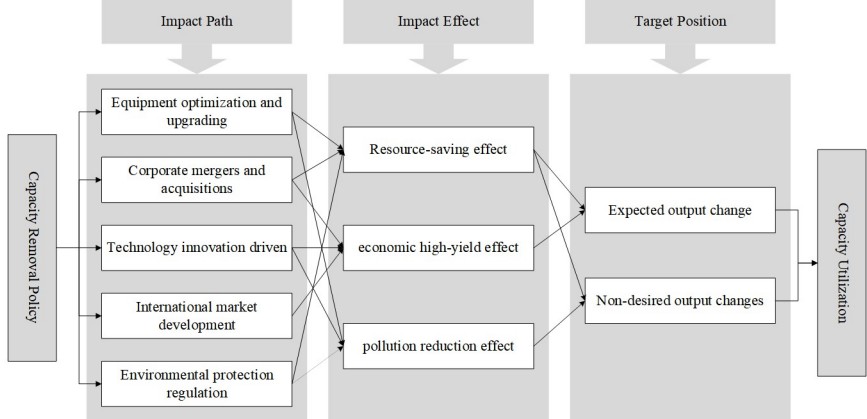

**Fig 1. The path framework of capacity utilization rate of steel industry affected by de-capacity policy.**

optimization and upgrading", "enterprise merger and reorganization", "technology innovation driving", "international market development", and "environmental protection regulation". These policies, in turn, affect changes in both expected and unexpected outputs in the steel industry, ultimately leading to improved utilization of production capacity in the steel industry. The following sections further elaborate on these five pathways.

Based on the analysis above, the following hypotheses are proposed:

- **Hypothesis 1:** The de-capacity policy directly and positively impacts the increase in the capacity utilization rate in the steel industry.

- **Hypothesis 2:** The de-capacity policy indirectly increases the capacity utilization rate in the steel industry through "equipment optimization and upgrading".

- **Hypothesis 3:** The de-capacity policy indirectly increases the capacity utilization rate in the steel industry through "enterprise mergers and restructuring".

- **Hypothesis 4:** The de-capacity policy indirectly increases the capacity utilization rate in the steel industry through "driving technological innovation".

- **Hypothesis 5:** The de-capacity policy indirectly increases the capacity utilization rate in the steel industry through "expanding international market access".

- **Hypothesis 6:** The de-capacity policy indirectly increases the capacity utilization rate in the steel industry through "environmental protection regulations".

## Path analysis

**Path 1: De-capacity policies → Equipment optimization and upgrading → Capacity utilization.** Supply-side structural reform in the context of the steel industry to production capacity policy document proposed "must be in accordance with the 'Industrial Structure Adjustment Guide Catalogue (2011) (Amendment)' of the relevant provisions, immediately shut down and dismantle 400 cubic meters and below iron blast furnace, 30 tons and below steelmaking converter, 30 tons and below steelmaking electric furnace and other backward production equipment" requirements. Equipment optimization and upgrading enhance capacity utilization by improving product quality, conserving resources, and mitigating pollution [31]. Firstly, advanced production equipment and processes can manufacture high-quality steel products, better aligning with market demand, resulting in increased market revenue and heightened capacity utilization. Secondly, in scenarios with fixed output levels, advanced production equipment and processes can reduce the consumption of energy, water, electricity, raw materials, and other production inputs, leading to lower production costs and cost minimization. Alternatively, when input levels are fixed, advanced production equipment and processes can augment the production of high-quality steel products, achieving maximum output. This results in profit maximization once the steel industry reaches producer equilibrium, consequently elevating capacity utilization. Finally, advanced production equipment and processes can curtail emissions of waste gas, wastewater, and solid waste, ultimately contributing to improved capacity utilization.

**Path 2: De-capacity policies → Enterprise mergers and restructuring → Capacity utilization.** In the context of supply-side structural reform, the policy on the removal of steel industry capacity proposed to "encourage qualified steel enterprises to implement cross-industry, cross-region, cross-ownership reduction and reorganization, and focus on promoting the implementation of mergers and reorganization of enterprises in large steel-producing provinces to exit some excess capacity". Mergers and acquisitions primarily enhance

capacity utilization by increasing industrial concentration and expelling underperforming enterprises from the market [32]. On one hand, mergers and acquisitions expedite the exit of underperforming enterprises from the market, effectively improving industry market concentration. High industry market concentration can lead to increased production profits and consequently boost the overall capacity utilization rate of the industry. On the other hand, the steel industry involves highly specialized production equipment. Through mergers and acquisitions, equipment devaluation losses can be minimized, while optimal capacity allocation can be achieved, thereby improving the capacity utilization rate of the merged and reorganized enterprises. This, in turn, enhances the capacity utilization rate of the industry as a whole.

**Path 3: De-capacity policies → Technology innovation driving → Capacity utilization.** In the context of supply-side structural reform, steel industry policy documents explicitly state the need to "enhance the level of intelligence in research and development, production, and services of enterprises and construct a group of intelligent manufacturing demonstration factories". In the era of high-quality development, technology innovation-driven initiatives serve as a key guarantee for companies to adapt to contemporary trends and bolster their market competitiveness. This plays a pivotal role in addressing overcapacity issues [33]. In the academic community, there are three differing perspectives on the impact of technology innovation-driven strategies on capacity utilization: Firstly, it is believed that technology innovation-driven approaches are favorable for improving capacity utilization. Secondly, it is suggested that technology innovation-driven strategies have a clear threshold effect on capacity utilization. Thirdly, it is noted that technology innovation-driven efforts have a distinct mediating effect within the path of capacity utilization concerning environmental regulations and openness to foreign markets [34, 35]. However, the overarching consensus is that the impact of technology innovation-driven strategies on capacity utilization is positive. They primarily influence capacity utilization through scale effects, structural effects, and mode effects. This is achieved by introducing new production equipment, developing new production processes to reduce production costs, decrease pollution emissions, increase marketing profits, and ultimately enhance capacity utilization.

**Path 4: De-capacity policies → International market development → Capacity utilization.** According to the theory of market supply and demand, it becomes challenging to balance the market under weak demand, leading to oversupply, increased inventories, a surge in excess capacity, and a decrease in capacity utilization. While the new round of anti-capacity policies predominantly focuses on the supply side to address excess capacity, there is still untapped potential for synergistic stimulation on the demand side, particularly through international market development [23]. In the context of supply-side structural reform, the policy paper on steel industry capacity reduction presents the idea of "encouraging enterprises with conditions to transfer part of their capacity through international capacity cooperation in conjunction with the construction of Belt and Road to achieve mutual benefits and win-win situation". International market development through trade and investment aims to reduce steel industry inventories and shift excess capacity, ultimately enhancing capacity utilization [36]. Throughout the process of trade and investment, enterprises enhance their production efficiency and address overcapacity by benefiting from knowledge spillover, technology spillover, market competition, and resource restructuring effects.

**Path 5: De-capacity policies → Environmental protection regulation → Capacity utilization.** The policy document for steel industry capacity reduction in the context of supply-side structural reform explicitly states that "the environmental protection law is strictly enforced, and continuous daily penalties are imposed on steel production capacity that fails

to meet the requirements of pollutant emissions, and in serious cases, the approval of the people's government with approval authority is reported, and the company is ordered to shut down or close down". Environmental protection regulations primarily drive enterprises to undertake technological innovation to enhance capacity utilization [37]. In the short term, environmental regulations compel non-compliant enterprises to halt production, which may not significantly address excess capacity [27]. However, in the long run, environmental protection regulations increase the cost of pollution for companies. For enterprises with high debt ratios, weak competitiveness, and outdated production methods, the risks associated with technological innovation, pollution fines, and administrative penalties are higher. These factors may prompt them to exit the market, thereby increasing industry concentration. Enterprises with comparative advantages must either innovate technologically or acquire new facilities and equipment to meet pollution emission standards. This renders traditional and outdated production equipment and processes obsolete, ultimately improving capacity utilization [38, 39].

## Research design

### Variable selection

**Explained variables.**   We selected the capacity utilization rate in the steel industry as the dependent variable, denoted as CU. In this case, we employed the Data Envelopment Analysis (DEA) model to measure the capacity utilization rate in the steel industry. Essentially, academically, the results obtained through this method are referred to as environmental technical efficiency. Fundamentally, it reflects the capacity and effort exerted by the steel industry in the pursuit of high-quality development. It serves as a comprehensive reflection of the influence of technological progress on economic development, resource utilization, and ecological impact, aligning with the contemporary requirements for high-quality development in the nation. In this study, a non-radial, non-angular SBM-DEA model was used to measure the capacity utilization of the Chinese steel industry. The calculated technical efficiency values were decomposed into pure technical efficiency and scale efficiency. We assumed that steel enterprises utilize N inputs to produce M desirable outputs and L undesirable outputs. The input vector is represented as $X = (X_1, X_2, X_N)$, the desirable output vector as $Y = (Y_1, Y_2, Y_N)$, and the undesirable output vector as $B = (B_1, B_2, B_N)$. Hence, the production technology of these enterprises can be defined within the following output set:

$$Q(X) = \{(Y, B): X can produce (Y, B)\} \quad (1)$$

The functional form of the Directional Distance Function for production technology is as follows:

$$D(X, Y, B; G_\gamma; -G_s) = MAX\{\rho : (Y + \rho G_\gamma, B - \rho G_s) \in Q(X)\} \quad (2)$$

Where, $G = (G_Y, G_B)$ represents the direction vector. This function explains the maximum desirable and minimum undesirable outputs under the existing technological conditions.

Considering the non-radial, non-angular expression of the Directional Distance Function, it is as follows:

$$\min \rho = \frac{1 - \frac{1}{N}\sum_{i=N}^{N} N \frac{S_i}{X_i^0}}{1 + \frac{1}{S_1+S_2}\left(\sum_{r=1}^{S_1} \frac{S_r^s}{Y_{t_0}^s} + \sum_{r=1}^{S_b} \frac{S_r^b}{Y_{t_0}^s}\right)} \tag{3}$$

$$X_0 = \lambda X + S^-$$

$$X_0 = \lambda X + S^+$$

$$Y_{r_0}^G = \lambda Y^G - S^G \tag{4}$$

$$Y_{r_0}^B = \lambda Y^G + S^B$$

$$\lambda \geq 0, S^- \geq 0, S^+ \geq 0, S^G \geq 0, S^B \geq 0$$

Where $\rho$ represents capacity utilization rate, $\lambda$ is the weight variable, $S^{+/-}$ represents input-output relaxation variable, and $N_i, S_i$ represents input type and output type.

Input indicators for capacity utilization models are typically chosen from a total factor perspective. However, limitations of the model and data availability often lead to the selection of representative production factors as input indicators. In literature that employs Data Envelopment Analysis to calculate capacity utilization, these input indicators frequently encompass three factors: capital stock, intermediate input, and labor input. Given data availability constraints, capital stock is typically represented by net fixed assets, intermediate input by energy consumption (in standard coal units), and labor input by the number of employees. The objective of the de-capacity policy in the steel industry is to enhance capacity utilization, lower costs, and boost profits. In this context, total profits are chosen as they effectively represent the anticipated output of the steel industry. Upon consulting the "China Iron and Steel Industry Yearbook", it was determined that the steel industry incurred an overall loss in 2015, resulting in negative total profits. However, Data Envelopment Analysis mandates that input-output indicators must be non-negative. Consequently, for model calculations, the total profits of the steel industry in 2015 were adjusted to 0. Steel production processes generate various pollutants, such as wastewater, smoke and dust, sulfur dioxide, carbon dioxide, and steel slag. Unfortunately, obtaining direct data on these emissions is hindered by limitations in statistical data. Consequently, carbon emissions are chosen as the undesirable output indicator. This selection aligns with the new national initiatives for energy conservation and emission reduction, reflecting the core of technological capacity. Regrettably, the "China Industrial Statistics Yearbook" and the "China Iron and Steel Industry Yearbook" do not include data on carbon emissions for the steel industry. Therefore, an estimation method involving coefficients is employed. Following the research methodology of Shangguan Qin [40], the final energy consumption (measured in standard coal units) for the black metal mining industry and the middle black metal smelting and rolling processing industry, as reported in the "China Energy Statistics Yearbook" is used as the source data for carbon emissions. Carbon emissions are estimated by combining this data with the carbon emission coefficient for standard coal (0.7599). The input-output indicators utilized for calculating the steel industry's capacity utilization are detailed in Table 1.

Fig 2 illustrates the changing trend in the capacity utilization of the steel industry from 2008 to 2019. It can be observed that the comprehensive efficiency of capacity utilization in the steel industry experienced two distinct phases. From 2008 to 2015, it went through a phase of

**Table 1. Input-output indicators for measuring capacity utilization in the steel industry using the data envelope model.**

| Input/output items | Characterization indicators | Data source and description |
|---|---|---|
| 3*Elemental inputs | Net value of fixed assets | China Iron and Steel Industry Yearbook |
| | Energy consumption (standard coal) | China Iron and Steel Industry Yearbook |
| | Number of industry employees | China Iron and Steel Industry Yearbook |
| Expected output | Total profit | China Iron and Steel Industry Yearbook |
| Non-desired output | Carbon Emissions | Obtained by formula calculation $C = FQ$; Where, C is carbon emission, F is coal carbon emission coefficient, and Q is standard coal consumption. |

fluctuating decline, with capacity utilization remaining below 40%, and even reaching 0% in 2015, once again confirming the view of low capacity utilization in China's steel industry. The period from 2015 to 2019 marked a phase of fluctuating increase, reaching its historical peak in 2018.

**Explanatory variables.**

1. De-capacity policy. Considering that the supply-side structural reform was formally proposed at the end of 2015, and the policy document "Opinions on Dissolving Excess Capacity in the Iron and Steel Industry to Achieve Destructive Development" was released in January 2016, 2015 was chosen as the policy shock time breakpoint, and the policy applied variable was set as "before 2015 = 0, after 2015 = 1", noted as Dcp.

2. **Equipment optimization and upgrading:** The "Opinions on the steel industry to resolve excess capacity to achieve development" on the iron and steel industry blast furnace, converter and electric furnace technical standards, and these three-production equipment in different parts of the steel production, the impact on the steel industry capacity utilization rate there was heterogeneity. Therefore, three indicators were selected to reflect the optimization and upgrading of equipment, namely the proportion of iron-making blast furnaces above 1000 m3, the proportion of converters above 100 tons and the proportion of electric furnaces above 30 tons, and the arithmetic average weighting method is applied to obtain a comprehensive index for characterization, which was recorded as Eor.

3. **Enterprise mergers and reorganizations:** Both the number of enterprises and industry concentration can reflect the mergers and reorganizations of enterprises. In view of the continuity and availability of data, the number of enterprises was selected for characterization and is denoted as Emr.

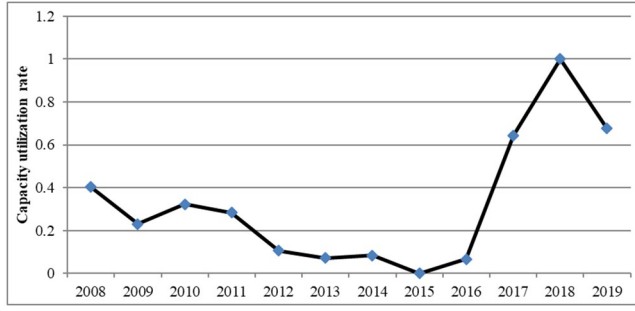

**Fig 2. Capacity utilization trend of steel industry from 2008 to 2019.**

4. **Technological innovation drive:** From the literature, it was found that technological innovation drive plays a mediating role in energy saving and emission reduction in overcapacity industries [33]. Therefore, three indicators, namely, integrated energy consumption per ton of steel, integrated emission rate of pollutants and integrated reuse rate of pollutants, were selected to reflect the technological innovation drive, and the same arithmetic average weighting method was adopted to obtain the composite index for characterization, which was denoted as Ti. Among them, the integrated reuse rate of pollutants was expressed as the average of three indicators, namely, reuse rate of wastewater, slag utilization rate and combustible gas recovery rate.

5. **International market development:** International market development channels are mainly international trade, international technology transfer and international investment and other forms. Considering that China's iron and steel enterprises mainly develop international markets through international trade, here, the proportion of exports of iron and steel products to business revenue was chosen for characterization, which is recorded as Imd.

6. **Environmental protection regulation:** According to the documents and important meetings issued by the Chinese government related to the removal of production capacity, ROST6.0 software was used to count the frequency of the words "steel industry", "removal of production capacity", "energy saving and emission reduction" and "pollution prevention" and "environmental protection inspector" in the documents and important meetings. The frequency of words such as "steel industry", "energy saving and emission reduction", "pollution prevention" and "environmental protection inspector" was characterized by the number of word frequency, which was recorded as Epr.

7. **Interactive variables:** The interaction terms of the de-capacity policy" and "equipment optimization and upgrading", "enterprise merger and reorganization", "technology innovation driving", "international market development"and "environmental protection regulation", which were recorded as Dcp×Eor, Dcp×Emr, Dcp×Ti, Dcp×Imd and Dcp×Epr, respectively.

**Control variables.** According to existing studies, four indicators, debt ratio (Dr), economic cycle (Fc), marketability level (Ml), and industry capital intensity (Icd), were selected as control variables [12, 41–43].

1. **Debt Ratio:** In this study, the debt ratio is represented as the ratio of current assets to long-term liabilities, denoted as Dr. The data is sourced from the "China Industrial Statistics Yearbook" for the years 2009 to 2019.

2. **Economic Cycle:** The measurement of the economic cycle utilizes the actual economic growth rate, denoted as Fc, as an indicator of economic fluctuations. Economic growth rate data is derived from the "China Statistical Yearbook" for the years 2009 to 2019.

3. **Marketization Level:** The marketization level in this paper is directly characterized using the marketization index compiled by Xiaolu Wang [44], denoted as Ml. This index is composed of a weighted combination of indicators representing market competition concentration in the steel industry and marketization of resource allocation, providing a holistic reflection of the marketization level within the steel industry.

4. **Industry Capital Density Index:** Capital intensity exhibits an inverse relationship with capacity utilization. In this paper, capital density is represented as the ratio of fixed asset

original value to labor input, denoted as Icd. Data for this index is sourced from the "China Labor Statistical Yearbook" for the years 2009 to 2019.

## Model construction

The academic community often employs double-difference models when studying policy effects [11, 26, 34]. However, typically, double-difference models are suitable for panel data, and the constructed experimental and control groups should exhibit clear boundaries and differences. While China has focused its efforts on resolving overcapacity issues in industries such as steel, coal, cement, glass, and electrolytic aluminum, overcapacity problems are also prevalent in various other industrial sectors. The capacity reduction policies have an impact on all industrial sectors, and resolving overcapacity in the steel industry can have significant spillover effects on upstream and downstream enterprises. Therefore, it is challenging to identify industries within the industrial sector that exhibit distinct differences from the steel industry. Even if such industries were identified, comprehensive panel data for those industries might not be readily available. Hence, in this study, we have opted for the OLS model for examination. To explore the impact pathways of the de-capacity policy on capacity utilization in the steel industry, and based on the aforementioned theoretical mechanisms, we construct the following econometric model:

$$CU_t = \delta_1 + \alpha_1 Dcp_t + \varepsilon_1 \tag{5}$$

Where, $CU_t$ is the capacity utilization rate of the steel industry, $Dcp_t$ is the variable of the de-capacity policy, $\delta_1$ is the constant term, $\alpha_1$ is the elastic coefficient, and $\varepsilon_1$ is the random disturbance term. This formula is used to test the relationship between the capacity reduction policy and the capacity utilization rate of the steel industry.

On the basis of formula (5), add control variables to test the relationship between capacity reduction policy and capacity utilization rate of steel industry:

$$CU_t = \delta_2 + \beta_1 Dcp_t + \beta_t Cv_t + \varepsilon_2 \tag{6}$$

Where, $CU_t$ is the capacity utilization rate of the steel industry, $Dcp_t$ is the variable of the capacity reduction policy, $Cv_t$ is the control variable (Dr, Icd, Fc, Ml), $\delta_2$ is the constant term, $\beta_1$ and $\beta_2$ is the elastic coefficient, and $\varepsilon_2$ is the random disturbance term.

On the basis of formula (6), explanatory variables and their interaction terms are added to test the action path of capacity reduction policy affecting the capacity utilization rate of the steel industry:

$$CU_t = \delta_3 + \omega_1 Dcp_t + \omega_t Cv_t + \omega_t^* Mv_t + \omega_t^{**} Iv_t + \varepsilon_3 \tag{7}$$

Where, $CU_t$ is the capacity utilization rate of the steel industry, $Dcp_t$ is the variable of the capacity reduction policy, $Cv_t$ is the control variable, $Mv_t$ is the explanatory variable (Eor, Emr, Ti, Imd, Epr), $Iv_t$ is the interaction term (Dcp×Eor Dcp×Emr Dcp×Ti Dcp×Imd Dcp×Epr), $\delta_3$ is the constant term, $\omega_1$, $\omega_t$, $\omega_t^*$, $\omega_t^{**}$ is the elastic coefficient, and $\varepsilon_3$ is the random disturbance term.

In order to mitigate the impact of heteroscedasticity on the model, empirical analysis was conducted using the natural logarithm of the aforementioned variables, excluding the de-capacity policy variable.

## Data sources

The data for the aforementioned variables primarily originated from sources including the "Steel Industry Yearbook", "China Industrial Yearbook", "China Energy Statistical Yearbook",

**Table 2. Descriptive statistics of variables.**

| Variable Name | Variable Symbol | Mean | Standard Deviation | Min | Max |
|---|---|---|---|---|---|
| Capacity Utilization Rate | CU | 0.31 | 0.33 | 0.00 | 1.00 |
| De-Capacity Policy | Dcp | 0.36 | 0.50 | 0.00 | 1.00 |
| Equipment Optimization and Upgrading | Eor | 43.42 | 7.67 | 33.40 | 55.32 |
| Enterprise Merger and Reorganization | Emr | 9789.09 | 2993.95 | 5113.00 | 14377.00 |
| Technological Innovation Drive | Ti | 96.77 | 1.43 | 94.42 | 99.18 |
| International Market Development | Imd | 0.88 | 0.22 | 0.64 | 1.43 |
| Environmental Protection Regulation | Epr | 58.27 | 22.72 | 23.00 | 90.00 |
| Debt Ratio | Dr | 0.66 | 0.03 | 0.61 | 0.71 |
| Capital Intensity | Icd | 92.58 | 41.70 | 40.99 | 141.61 |
| Economic Cycle | Fc | 10.86 | 3.93 | 7.04 | 18.40 |
| Marketization Level | Ml | 5.98 | 0.67 | 5.07 | 7.04 |

and "China Statistical Yearbook" for the period spanning from 2009 to 2019. In certain instances and for specific indicators, missing data were substituted with information from industry reports and relevant literature, or alternatively, interpolation methods were applied for data completion, as detailed in Section "Variable selection". Descriptive statistics for the variables are presented in Table 2, and the model estimation was conducted using EViews software.

## Analysis of results

### Baseline regression analysis

Model 1 examines the relationship between the de-capacity policy and capacity utilization rate independently. Model 2, building upon Model 1, incorporates control variables to investigate the relationship between the de-capacity policy and capacity utilization rate. Table 3 presents the regression results of the impact of the de-capacity policy on the steel industry.

Model 1 presents the regression results of the impact of the de-capacity policy on the capacity utilization in the steel industry. The results indicate that the variable Dcp passes significance tests at least at the 10% level and has a positive coefficient. This suggests that the de-capacity policy has played a positive role in enhancing the capacity utilization in the steel industry, confirming Hypothesis 1. Since the introduction of the Supply-Side Structural Reform in 2015, the document titled "Opinions on Resolving Overcapacity in the Steel Industry for

**Table 3. Test results of capacity utilization rate of steel industry affected by de-capacity policy.**

| Variables | Model 1 | Model 2 |
|---|---|---|
| **Dcp** | 0.454** (2.905) | 0.027* (2.560) |
| **Ctr** | | -5.400* (-2.507) |
| **Icd** | | -0.005* (-2.679) |
| **Fc** | | 0.035 (0.608) |
| **Ml** | | 0.148 (0.579) |
| $R^2$ | 0.484 | 0.850 |
| **P-value** | 0.017 | 0.068 |

Note: "*" denotes 10% significance level, "**" denotes 5% significance level, and "****" denotes 1% significance level. () within indicates t-value.

Transformation and Development" was subsequently released. The Chinese government has successively introduced related supporting policy documents. Under the macroeconomic regulation and control within this series of policy systems, the steel industry has achieved unprecedented success in reducing overcapacity. By the end of 2018, China's steel industry had eliminated 150 million tons of outdated capacity, and 64.74 million tons of crude steel capacity from "zombie enterprises" had been retired, surpassing the targets set for capacity reduction in the 13th Five-Year Plan.

Model 2 incorporates control variables to examine the relationship between the de-capacity policy and the capacity utilization in the steel industry. From the estimation results, it is evident that the debt ratio significantly negatively affects capacity utilization. A high debt ratio is a typical characteristic of numerous zombie enterprises in the steel industry, severely constraining the production and operational capabilities of these enterprises. In adverse market conditions, this leads to substantial losses and hinders the improvement of capacity utilization. Capital density also significantly negatively affects capacity utilization. A higher capital density implies that labor possesses more production equipment, increasing the likelihood of overcapacity and consequently restricting the improvement of capacity utilization. The economic cycle and marketization level do not exhibit significant effects on capacity utilization.

## Impact path analysis

Building upon the foundation laid out in the previous sections, Table 4 introduces explanatory variables and their interaction terms to examine the pathways through which the de-capacity policy affects capacity utilization in the steel industry. Models 3 to 7 extend Model 2 by including variables such as equipment optimization and upgrading, enterprise merger and reorganization, technology innovation driving, national market development, and environmental protection regulation, as well as their interaction with the de-capacity policy, to assess the pathways through which the de-capacity policy affects capacity utilization rate.

**Table 4. The path test results of de-capacity policy affecting capacity utilization rate of steel industry.**

| Variables | Model 3 | Model 4 | Model 5 | Model 6 | Model 7 |
|---|---|---|---|---|---|
| Dcp | 0.001* (2.869) | 0.005* (2.777) | 0.203*** (5.069) | 0.086 (1.560) | 0.313*** (6.570) |
| Eor | 0.131* (2.560) | | | | |
| Emr | | -0.250** (3.568) | | | |
| Ti | | | 0.741*** (6.001) | | |
| Imd | | | | -0.062 (-0.034) | |
| Epr | | | | | 0.453** (3.364) |
| Ctr | -0.511* (-2.888) | -0.518** (-3.529) | -0.272** (-3.560) | -0.559* (-1.993) | -0.639 (-0.861) |
| Icd | -0.805** (-3.960) | -0.560* (-2.331) | -0.338* (-2.570) | -0.747** (-3.465) | 0.317 (0.560) |
| Fc | 0.456 (0.560) | 0.411 (1.111) | 0.788 (1.006) | 0.445 (0.569) | 0.511 (0.060) |
| Ml | 0.313 (1.869) | 0.265* (2.881) | 0.349 (1.260) | 0.332 (1.863) | 0.280 (1.160) |
| Dcp×Eor | 0.068* (3.064) | | | | |
| Dcp×Emr | | -0.111* (2.775) | | | |
| Dcp×Ti | | | 0.487*** (5.571) | | |
| Dcp×Imd | | | | -0.008 (-1.264) | |
| Dcp×Epr | | | | | 0.408** (3.336) |
| $R^2$ | 0.746 | 0.849 | 0.901 | 0.847 | 0.806 |
| P-value | 0.094 | 0.090 | 0.016 | 0.089 | 0.061 |

Note: "*" denotes 10% significance level, "**" denotes 5% significance level, and "***" denotes 1% significance level. () within indicates t-value.

Model 3 presents the analysis of the results for Path 1. The outcomes indicate that Eor and Dcp×Eor have a significant positive impact on capacity utilization, both passing the significance test at a 10% level. This suggests that "equipment optimization and upgrades" represent a pathway through which the de-capacity policy positively affects capacity utilization, confirming Hypothesis 2. The correlation test confirms that the elasticity coefficient of the de-capacity policy with "equipment optimization and upgrades" is 0.517 (P < 10%), substantiating its positive role in promoting equipment optimization and upgrades in the steel industry.

Model 4 examines the results for Path 2. The findings reveal that Emr and Dcp×Emr significantly and positively impact capacity utilization, passing significance tests at 5% and 10% levels, respectively. This indicates that "business mergers and restructuring" represent a pathway through which the de-capacity policy positively affects capacity utilization, confirming Hypothesis 3. The correlation test shows that the elasticity coefficient of the de-capacity policy with "business mergers and restructuring" is -0.740 (P < 1%). After the implementation of the de-capacity policy, the steel industry witnessed an acceleration of mergers and restructurings, resulting in the removal of "zombie enterprises" and a decrease in the number of large-scale enterprises. This improvement favored the enhancement of industry concentration and resource allocation efficiency, yielding greater economies of scale and a positive impact on capacity utilization.

Model 5 analyzes the results for Path 3. The outcomes indicate that Ti and Dcp×Ti have a significantly positive impact on capacity utilization, both passing the 1% significance test. This suggests that "technology innovation-driven" represents a pathway through which the de-capacity policy positively affects capacity utilization, confirming Hypothesis 4. The correlation test demonstrates that the elasticity coefficient of the de-capacity policy with "technology innovation-driven" is 0.652 (P < 5%). After the implementation of the de-capacity policy, the acceleration of technology innovation drove efficient resource utilization and pollution reduction, positively impacting capacity utilization in the steel industry.

Model 6 evaluates the results for Path 4. Imd and Dcp×Imd negatively influence capacity utilization but do not pass the statistical test, Hypothesis 5 is not verified. This implies that the potential for improving capacity utilization in the steel industry through stimulating export trade via demand-side measures is limited, confirming the shift in China's approach to resolving overcapacity from the supply side.

Model 7 presents the analysis of results for Path 5. Epr and Dcp×Epr significantly and positively impact capacity utilization, both passing significance tests at 5% levels. This suggests that "environmental protection regulations" represent a pathway through which the de-capacity policy positively affects capacity utilization, confirming Hypothesis 6. The correlation test shows that the elasticity coefficient of the de-capacity policy with "environmental protection regulations" is 0.542 (P < 10%). After the introduction of the "Opinions on Resolving Overcapacity in the Steel Industry for Transformation and Development" document in 2016, led by the concept of green development, relevant national departments strengthened environmental regulations. This not only outlined specific requirements for environmental protection in policy documents related to capacity reduction but also introduced specific regulations in environmental protection. A series of environmental protection regulations compelled companies to engage in technological innovation, forced zombie enterprises out of the market, and had a positive impact on enhancing capacity utilization in the steel industry.

Through these models, it is evident that the de-capacity policy positively correlates with capacity utilization in the steel industry. The policy achieves this through four significant pathways: "equipment optimization and upgrades", "business mergers and restructuring", "technology innovation-driven", and "environmental protection regulations". The interaction effects of the de-capacity policy with "technology innovation-driven" and "environmental

protection regulations" are notably stronger than their interaction with "equipment optimization and upgrades" and "business mergers and restructuring". This signifies that under the backdrop of supply-side structural reform, the de-capacity policy primarily affects capacity utilization in the steel industry through "technology innovation-driven" and "environmental protection regulations".

## Robustness test

The first method is the placebo test. In the previous section, 2015 was chosen as the time breakpoint for the implementation of the de-capacity policy. Now, let's assume that the policy breakpoint is not in 2015 but in other years. Specifically, in this case, the empirical process will be repeated with the policy breakpoint placed one year before and one year after, as shown in Table 5. Using 2014 as the policy breakpoint to test the impact of the five pathways on capacity utilization results in Models 8–12. In the test models, none of the variables is significant, and all models have relatively low goodness of fit and explanatory power. Using 2016 as the policy

**Table 5. Test results for setting other policy breakpoint times.**

| Variable | Policy break Time (2014) | | | | | Policy break Time (2014) | | | | |
|---|---|---|---|---|---|---|---|---|---|---|
| | Model 8 | Model 9 | Model 10 | Model 11 | Model 12 | Model 13 | Model 14 | Model 15 | Model 16 | Model 17 |
| Dcp | 0.020 | 0.124 | 0.343 | 0.062 | 0.110 | 0.129* | 0.222 | 0.557** | 0.029 | 0.133 |
| | 0.694 | 0.896 | 1.254 | 1.650 | 2.004 | 2.551 | 0.691 | 3.794 | 0.653 | 1.678 |
| Eor | 0.109 | | | | | 0.228 | | | | |
| | 1.992 | | | | | 1.992 | | | | |
| Emr | | -0.146 | | | | | -0.204 | | | |
| | | -1.647 | | | | | 2.241 | | | |
| Ti | | | 0.225 | | | | | 0.225 | | |
| | | | 2.046 | | | | | 2.046 | | |
| Imd | | | | -0.072 | | | | | -0.125 | |
| | | | | -1.146 | | | | | -1.155 | |
| Epr | | | | | 0.276 | | | | | 0.222* |
| | | | | | 0.674 | | | | | 0.476 |
| Ctr | -0.315 | -0.405* | -0.156** | -0.423* | -0.516 | -0.015* | -0.005* | -0.153** | -0.443 | -0.225 |
| | -2.556 | -4.329 | -3.770 | -2.365 | -0.861 | -1.556 | -2.329 | -3.780 | -1.365 | -0.361 |
| Icd | -0.867** | -0.540 | -0.348* | -0.558 | 0.328 | -0.760* | -0.540* | -0.368* | -0.758* | 0.347 |
| | -3.772 | -2.438 | -2.356 | -3.439 | 1.372 | -2.732 | -2.334 | -2.230 | -3.499 | 1.362 |
| Fc | 0.476 | 0.416 | 0.774 | 0.455 | 0.541 | 0.446 | 0.456 | 0.710 | 0.350 | 0.549 |
| | 0.477 | 1.170 | 1.348 | 0.628 | 0.266 | 0.414 | 1.576 | 1.344 | 0.646 | 0.232 |
| Ml | 0.513 | 0.137* | 0.249 | 0.358 | 0.253 | 0.539 | 0.187* | 0.249 | 0.387 | 0.263 |
| | 1.367 | 2.453 | 1.256 | 1.358 | 1.136 | 1.364 | 2.458 | 1.266 | 1.390 | 1.139 |
| Dcp×Eor | 0.054 | | | | | 0.851 | | | | |
| | 1.396 | | | | | 1.394 | | | | |
| Dcp×Emr | | -0.048 | | | | | -0.334 | | | |
| | | -1.687 | | | | | -1.687 | | | |
| Dcp×Ti | | | 0.277 | | | | | 0.172 | | |
| | | | 1.689 | | | | | 3.485 | | |
| Dcp×Imd | | | | -0.100 | | | | | -0.161 | |
| | | | | -0.687 | | | | | -0.288 | |
| Dcp×Epr | | | | | 0.400 | | | | | 0.203 |
| | | | | | 1.421 | | | | | 1.181 |
| $R^2$ | 0.638 | 0.550 | 0.646 | 0.756 | 0.619 | 0.550 | 0.641 | 0.660 | 0.612 | 0.693 |
| P-value | 0.345 | 0.138 | 0.224 | 0.190 | 0.216 | 0.089 | 0.166 | 0.099 | 0.766 | 0.124 |

**Note:** "*" denotes 10% significance level, "**" denotes 5% significance level, and "***" denotes 1% significance level. () within indicates t-value.

breakpoint to test the impact of the five pathways on capacity utilization results in Models 13–17. In these test models, variables Dcp, Dcp, and Epr are significant at least at the 10% significance level, while all other variables are not significant. All models have relatively low goodness of fit and explanatory power. Overall, when testing the impact of the de-capacity policy on the steel industry's capacity utilization with 2014 and 2016 as policy time breakpoints, the model's performance (variables passing significance tests is very limited, and the goodness of fit of the models is relatively low) is poor. This indicates that the research results are robust when choosing 2015 as the policy time breakpoint.

The second approach involves changing the dependent variable. In the preceding analysis, the capacity utilization rate of the steel industry, considering non-desired output, was computed. Now, the empirical process is retested using the capacity utilization rate of the steel industry as the dependent variable without considering non-desired output. This process results in Models 18 to 23, and the test results, as depicted in Table 6, reveal that the variable signs remain consistent with the estimates in Table 3, affirming the robustness of the results.

**Table 6. Test results that do not take into account undesirable outputs.**

| Variable | Model 18 | Model 19 | Model 20 | Model 21 | Model 22 | Model 23 |
|---|---|---|---|---|---|---|
| Dcp | 0.200*<br>2.882 | 0.151*<br>2.869 | 0.017*<br>2.347 | 0.227**<br>4.162 | 0.116<br>1.060 | 0.240***<br>4.550 |
| Eor | | 0.229*<br>2.586 | | | | |
| Emr | | | -0.133**<br>2.998 | | | |
| Ti | | | | 0.571***<br>4.018 | | |
| Imd | | | | | -0.102<br>-0.114 | |
| Epr | | | | | | 0.444**<br>3.550 |
| Ctr | -2.403*<br>-2.266 | -0.420*<br>-2.234 | -0.415**<br>-4.029 | -0.142**<br>-3.575 | -0.326*<br>-2.400 | -0.512<br>-0.711 |
| Icd | -0.045*<br>-2.544 | -0.814*<br>-3.536 | -0.830*<br>-2.252 | -0.208*<br>-2.386 | -0.550**<br>-3.300 | 0.308<br>1.432 |
| Fc | 0.021<br>0.638 | 0.499<br>0.360 | 0.411<br>1.167 | 0.744<br>1.320 | 0.425<br>0.522 | 0.511<br>0.060 |
| Ml | 0.124<br>0.879 | 0.511<br>1.339 | 0.168*<br>2.186 | 0.231<br>1.110 | 0.311<br>1.723 | 0.242<br>1.051 |
| Dcp×Eor | | 0.116*<br>2.465 | | | | |
| Dcp×Emr | | | -0.101*<br>2.795 | | | |
| Dcp×Ti | | | | 0.586***<br>5.661 | | |
| Dcp×Imd | | | | | -0.028<br>-1.336 | |
| Dcp×Epr | | | | | | 0.348**<br>4.345 |
| $R^2$ | 0.520 | 0.767 | 0.777 | 0.834 | 0.859 | 0.720 |
| P-value | 0.038 | 0.072 | 0.099 | 0.067 | 0.091 | 0.112 |

**Note:** "*" denotes 10% significance level, "**" denotes 5% significance level, and "***" denotes 1% significance level. () within indicates t-value.

## Conclusions

This paper empirically examines the impact and pathways of capacity reduction policies on the capacity utilization in the iron and steel industry using time-series data from 2008 to 2019. The study indicates that capacity reduction policies have significantly improved the capacity utilization in the steel industry, and this improvement is achieved through the combined effects of "equipment optimization and upgrade", "enterprise mergers and restructuring", "technology innovation-driven", and "environmental protection regulations". Among these pathways, "technology innovation-driven" and "environmental protection regulations" play dominant roles, while the "international market expansion" pathway did not exhibit its expected impact. To ensure the continued effectiveness of capacity reduction policies, the government should not only strengthen the "technology innovation-driven" and "environmental protection regulations" pathways but also activate the "national market expansion" pathway, fully exploring the potential of international cooperation to enhance the capacity utilization in the steel industry.

As time progresses, policy effects may experience a diminishing trend. This paper provides a scientific theoretical foundation for how the government can further improve capacity utilization in the steel industry in the future and can serve as a reference for other countries facing similar overcapacity issues. However, this paper's assessment of the performance of capacity reduction policies is not comprehensive. The impact of these policies is multifaceted, and this paper only examines their influence on capacity utilization in the steel industry. Subsequent research could construct a more comprehensive policy evaluation system that takes into account employment, industry structure optimization, fiscal revenue, economic output, and various other factors.

## Supporting information

**S1 Data.**
(DOCX)

## Author Contributions

**Data curation:** Hongsong Tang.

**Investigation:** Hongsong Tang.

**Methodology:** Yang Liu.

**Writing – original draft:** Shixin Shi, Hao Li.

**Writing – review & editing:** Hao Li.

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
