## [Decision Letter · Decision Letter 0]

6 Oct 2023

PONE-D-23-28251A study of the impact of de-capacity policies on industry capacity utilization paths: Evidence from the Chinese steel industryPLOS ONE

Dear Dr. Li,

Thank you for submitting your manuscript to PLOS ONE. After careful consideration, we feel that it has merit but does not fully meet PLOS ONE’s publication criteria as it currently stands. Therefore, we invite you to submit a revised version of the manuscript that addresses the points raised during the review process. Changes which must be made before publication:Please restructure the abstract and make sure to present the findings in a clear fashion.Emphasize your contributions by comparing your work to other published work in the fieldAdd a limitations section.The LR needs to be enhanced further by exploring international studies to clarify deficiencies in the current body of literature.Please submit your revised manuscript by Nov 20 2023 11:59PM. If you will need more time than this to complete your revisions, please reply to this message or contact the journal office at plosone@plos.org. Please include the following items when submitting your revised manuscript:

We look forward to receiving your revised manuscript.

Kind regards,

Rateb J. Sweis

Academic Editor

PLOS ONE

Journal Requirements:

Additional Editor Comments:

Manuscript ID PONE-D-23-28251 entitled "A study of the impact of de-capacity policies on industry capacity utilization paths:

Evidence from the Chinese steel industry" which you submitted to PLOS ONE, has been reviewed. The comments of the reviewers are included at the bottom of this letter.

Four of the reviewers have recommended your paper for publication, subject to major revisions while one reviewer gave you minor revisions. Therefore, we invite you to respond to the reviewers' comments and revise your manuscript. Please address the concerns of each reviewers in a clear and concise manner

Reviewers' comments:

Reviewer's Responses to Questions

**Comments to the Author**

1. Is the manuscript technically sound, and do the data support the conclusions?

Reviewer #1: No

Reviewer #2: Yes

Reviewer #3: Yes

Reviewer #4: Partly

Reviewer #5: Yes

2. Has the statistical analysis been performed appropriately and rigorously? 

Reviewer #1: No

Reviewer #2: Yes

Reviewer #3: Yes

Reviewer #4: No

Reviewer #5: No

3. Have the authors made all data underlying the findings in their manuscript fully available?

Reviewer #1: Yes

Reviewer #2: Yes

Reviewer #3: Yes

Reviewer #4: Yes

Reviewer #5: Yes

4. Is the manuscript presented in an intelligible fashion and written in standard English?

Reviewer #1: No

Reviewer #2: Yes

Reviewer #3: Yes

Reviewer #4: No

Reviewer #5: No

5. Review Comments to the Author

Reviewer #1: The abstract should present the study's findings in different wording, and personal pronouns should be avoided.

The study effectively elucidates the background problem, but it falls short in clarifying the deficiencies present in the current body of literature. Consequently, it is advisable for the authors to emphasize the likely knowledge gaps and integrate these into their study results.

The analytical framework and data envelope model should be made clearer to reflect the subject of the study and have in-depth bases from the literature.

A more comprehensive explanation of Figure 1 is needed, encompassing all the used components and their interconnections.

The examination of Paths should be guided by the knowledge present in the existing literature.

The title "Explanatory Variables" was duplicated in both sections 3.1.1 and 3.1.2. To address this, the authors can provide an introduction of the components of the Explanatory Variables and adjust the numbering of the titles accordingly.

Section the 2.1.3 needs to be edited more clearly.

The authors ought to provide a clearer explanation for the justification behind their choice of (Debt ratio, economic cycle, marketability level, and industry capital intensity) as control variables.

The study presented the outcomes of models 1 through 7 without a substantive discussion regarding the methodology employed to formulate these models.

The study does not adequately emphasize the notable contributions it makes in comparison to prior research. Also, it lacks a section addressing its limitations and potential avenues for future research.

The level of language is adequate but it should be proofread before re-submission. Grammar should be revised before resubmitting the paper since several errors can be found in the text, especially with regard to prepositions, punctuation marks, and tenses. Also, avoid using personal pronouns in the academic paper.

In the question section, the authors filled in the following (The authors received no specific funding for this work) while at the end of the study, the authors wrote (Funding: The National Social Science Foundation Project of China, Grant/Award Numbers: 17XJY014). Please clarify this point.

Reviewer #2: Positive Aspects:

- The article is well organized and provides valuable insights about research topic in the field. Moreover, the structure of the analysis is clear and logical.

- Research design is clear, comprehensive, and detailed. Also, the selection of variables is well justified and the integration between them and other factors enriches the analysis by explicitly examining the policy's effects on capacity utilization.

- Conclusions and implications provide a clear summary of the study's findings and offer actionable recommendations for policymakers.

Suggestions for Improvements:

- Literature Review Section:

- Adding more examples can enhance the research. E.g., the researchers can incorporate some practical examples or case studies from China and/or other real-world examples to improve the research practicality and relevance. These of course should be recent examples and cases.

- Research Design:

- Clarifying methodologies. For example, the use of a data envelope model is mentioned for measuring the capacity utilization rate. Therefore, providing more explanations and references regarding this model and how it's applied would enhance the clarity of the research design.

- Expanding on Data Sources: For example, providing more information on the reliability, representativeness, and accuracy of the data to enhance the research design.

- Discussing Limitations: For example, addressing potential limitations of the chosen methodologies, data sources, and the overall research design should add transparency and ensure a vigorous interpretation of results.

- Conclusions:

- Incorporating Quantitative Results: Incorporating specific quantitative results or statistics from the analyzed tests can enhance the credibility and clarity of the conclusions regarding the vitality of the chosen policy time breakpoint.

- Providing Citations for Theoretical Analysis: Providing references or citations to support the theoretical assertions can strengthen the argument and enhance academic and research rigor.

- Elaborating more on Industry Cultivation Mechanism and multi-dimensional collaborative supervision: One of the research recommendations is that the need for improving the industry cultivation mechanism and multi-dimensional collaborative supervision. Expanding on what this entails and how it can positively impact capacity utilization would provide more clarity.

- Limitations:

- Addressing Potential Limitations: Such as assumptions or constraints, to ensure a balanced view and promote transparency in the research.

- Grammar and Style:

- There are minor grammatical errors. Stylistic improvements can be made to enhance the overall flow and readability of the text.

Reviewer #3: In general, this is a well-written journal article. The authors have selected the "de-capacity" policy executed in China since 2010s, which is a topic that worth of discussing.

However, some minor revisions should be made to extend the depth of this paper:

1. Please try to add some content on the Introduction and Literature reivew.

2. In 2.2, please tell the readers the significance and the linkages of each path analysis.

3. Please re-check the fonts of the equations.

4. Please tell the research deficiencies & future works in the conclusion part.

5. Finally, please re-check the grammar, English expression and other details of the whole paper.

Reviewer #4: The paper presents some interesting results, but it suffers from a lack of clarity and coherence. The objectives of the paper are not well defined and the flow of the paper is confusing. The paper would benefit from a thorough revision of the structure and the language. There are also many typographical errors that need to be corrected.

Reviewer #5: abstract needs an introductory statement. the author starts with data data duration and discussing the model.

methodology is not clear in abstract

results of the work should be clear in the abstract as well

literature Review discusses the issue of over capacity and suggested solutions but only for China. it is clear that the data is for China but there should be literature for other countries and what other countries suggested as solutions to this problems' which the authors can reflect on in their proposed solutions.

equations 1 to 3 parameters' are not clear the letters are tangled please fix this formation issue please, what is w* ** , alpha, beta and omega what do you mean by elasticity coefficients?

for the same suggested model equations 1 to 3. the paragraph after it describes the model parameters in the past tense. it should be in present tense

the whole model explanation is not clear and needs to be rewritten in clear way

section 3.3: refer to those references in the table, yet again those steel industry year books are they public information and can they be listed in the references? even if data is interpolated you should indicate that in the table data

sec. 4.1 you start by explaining table 3 but how did the 7 models were obtained? and why are they considered models it is not clear? which equations 1 to 3 are used, give example to that for the data in table 3.

after verifying the models listed, the data analysis is also better explained in plots. the section is long and with large paragraphs and needs major rewriting.

sec. 4.2: start by explaining the fist method placebo test, was this listed in the methods? and how was it used for years other than 2015 as other years are mentioned. later in the same long paragraph it is mentioned that 2014 and 2016 only were used this requires to explain why only those years and it is worth it to see the effect of the proceeding years.

conclusion is too long and in one paragraph

6. PLOS authors have the option to publish the peer review history of their article (what does this mean?). If published, this will include your full peer review and any attached files.

Reviewer #1: No

Reviewer #2: No

Reviewer #3: **Yes: **Penghao YE

Reviewer #4: No

Reviewer #5: No

---

## [Author Response · Author response to Decision Letter 0]

10 Nov 2023

Dear Editors and Reviewers,

We would like to thank you for your efforts in reviewing our manuscript titled “A study of the impact of de-capacity policies on industry capacity utilization paths: Evidence from the Chinese steel industry”. We greatly appreciate your acknowledgment of our work and for providing us with an opportunity to revise our manuscript. At the same time, we would like to express our regret for not meeting the quality standards of the PLOS ONE journal. We have studied editors’ and reviewers’ comments carefully. According to the detailed suggestions, we have made a careful revision on the original manuscript. All revised portions are marked in yellow in the revised manuscript which we would like to submit for your kind consideration. Additionally, clean revised manuscript and supporting information are also uploaded.

The main corrections and detailed responses to comments in this article can be found in the document 'Response to Reviewers'.

Thank you again for your positive comments and valuable suggestions on improving the quality of our manuscripts.

Sincerely,

Hao Li

202121000581@stu.swpu.edu.cn

---

## [Decision Letter · Decision Letter 1]

27 Nov 2023

A study of the impact of de-capacity policies on industry capacity utilization paths: Evidence from the Chinese steel industry

PONE-D-23-28251R1

Dear Dr. Li,

We’re pleased to inform you that your manuscript has been judged scientifically suitable for publication and will be formally accepted for publication once it meets all outstanding technical requirements.

Kind regards,

Rateb J. Sweis

Academic Editor

PLOS ONE

Additional Editor Comments (optional):

Reviewers' comments:

Reviewer's Responses to Questions

**Comments to the Author**

1. If the authors have adequately addressed your comments raised in a previous round of review and you feel that this manuscript is now acceptable for publication, you may indicate that here to bypass the “Comments to the Author” section, enter your conflict of interest statement in the “Confidential to Editor” section, and submit your "Accept" recommendation.

Reviewer #3: All comments have been addressed

Reviewer #5: All comments have been addressed

2. Is the manuscript technically sound, and do the data support the conclusions?

Reviewer #3: Yes

Reviewer #5: Yes

3. Has the statistical analysis been performed appropriately and rigorously? 

Reviewer #3: Yes

Reviewer #5: N/A

4. Have the authors made all data underlying the findings in their manuscript fully available?

Reviewer #3: Yes

Reviewer #5: Yes

5. Is the manuscript presented in an intelligible fashion and written in standard English?

Reviewer #3: Yes

Reviewer #5: Yes

6. Review Comments to the Author

Reviewer #3: After the detailed 1st-round review & modification, the review comments have been seriously considered by the author(s). The manuscript R1 now shows sound topic expression, academic writing, logical essay structure, and adequate quantitative methods. Most of the deficiencies have been solved, thereafter, I personally do not have any more comments for the manuscript R1.

Summing up the above, my personal recommendation is ACCEPT. But before submitting the R2 version, please re-check carefully on the manuscript writing, grammar, expressions, and possible omitted interpretations of the research topics, method descriptions and results discussion.

Reviewer #5: all comments have been addressed and answered and changes made to the original manuals are clear and adhere to PLOS policy

7. PLOS authors have the option to publish the peer review history of their article (what does this mean?). If published, this will include your full peer review and any attached files.

Reviewer #3: **Yes: **Penghao YE

Reviewer #5: No

---

## [Editor Report · Acceptance letter]

8 Dec 2023

PONE-D-23-28251R1 

A study of the impact of de-capacity policies on industry capacity utilization paths: Evidence from the Chinese steel industry 

Dear Dr. Li:

I'm pleased to inform you that your manuscript has been deemed suitable for publication in PLOS ONE. Congratulations! Your manuscript is now with our production department. 

Kind regards, 

on behalf of

prof. Rateb J. Sweis 

Academic Editor

PLOS ONE